# Synthetic molecular evolution of hybrid cell penetrating peptides

W. Berkeley Kauffman[1], Shantanu Guha[1] & William C. Wimley[1]

Peptides and analogs such as peptide nucleic acids (PNA) are promising tools and therapeutics, but the cell membrane remains a barrier to intracellular targets. Conjugation to classical cell penetrating peptides (CPPs) such as $pTat_{48-60}$ (tat) and $pAntp_{43-68}$ (penetratin) facilitates delivery; however, efficiencies are low. Lack of explicit design principles hinders rational improvement. Here, we use synthetic molecular evolution (SME) to identify gain-of-function CPPs with dramatically improved ability to deliver cargoes to cells at low concentration. A CPP library containing 8192 tat/penetratin hybrid peptides coupled to an 18-residue PNA is screened using the HeLa pTRE-LucIVS2 splice correction reporter system. The daughter CPPs identified are one to two orders of magnitude more efficient than the parent sequences at delivery of PNA, and also deliver a dye cargo and an anionic peptide cargo. The significant increase in performance following a single iteration of SME demonstrates the power of this approach to peptide sequence optimization.

---

[1] Department of Biochemistry and Molecular Biology, Tulane University School of Medicine, 1430 Tulane Avenue, New Orleans, LA 70112, USA. Correspondence and requests for materials should be addressed to W.C.W. (email: wwimley@tulane.edu)

Peptides and peptide-like molecules are generating increasing interest as biotech tools and therapeutic agents[1, 2]. There are currently 60+ FDA-approved peptide drugs in the market with another 140+ in clinical trials and 500+ in pre-clinical development. The majority of approved peptides have extracellular targets because the cell membrane represents a barrier to intracellular targeting[3]. Similarly, antisense DNA analogs, including peptide nucleic acids (PNA) and phosphorodiamidate morpholino oligomers (PMOs), are generating growing excitement[3–5], but have yet to fully overcome limitations in the efficiency of delivery to the nuclei of the desired cells. Toward solving the delivery problem inherent to peptide, PNA, and PMO cargoes, cell penetrating peptides (CPPs) have shown promise as vehicles capable of transporting such cell-impermeant cargo to cytosolic or nuclear targets. However, there remains a need to identify CPPs with higher efficiencies, lower effective treatment concentrations, decreased cytotoxicity, and alternative mechanisms of action[2, 6–8].

Despite the need for improved CPPs, rational design is challenging due to the lack of explicit sequence–structure–function relationship rules[9]. In this work, we identify gain-of-function CPPs with useful properties using synthetic molecular evolution (SME). SME is an iterative process of designing rational combinatorial libraries that explore the sequence space around known templates, and screening such iterative libraries, orthogonally, to find members that display gain-of-function. It enables the utilization of known information, and the simultaneous testing of multiple hypotheses by rationally introducing constrained amino acid variability at specific locations throughout a template sequence. Previously, we have used SME to identify potent β-sheet pore-forming peptides[10–12], enhancers of receptor tyrosine kinase activation[13], spontaneous membrane translocating peptides[14], gain-of-function and loss-of-function pore-forming peptides[15, 16], pH-triggered pore-forming peptides[17], and antimicrobial peptides[18].

SME is used here to identify CPP sequences capable of efficiently delivering PNA, peptides, and other cargoes to living cells. PNAs are synthetic nucleic acid analogs possessing a peptide bond linked N-(2-aminoethyl) glycine backbone with the nucleobase (A/T/C/G) attached via methylene carbonyl bonds[19]. These molecules are resistant to enzymatic degradation, stably bind complimentary nucleic acids with high specificity and high affinity using Watson–Crick base pairing, and are easily conjugated to amino acid sequences because they share backbone chemistry. As antisense gene therapy agents, PNAs can modulate transcription, translation, splicing, and replication[20–27]. In this work, we screened for CPPs capable of nuclear delivery of PNA705, an 18-residue PNA sequence that sterically blocks an aberrant splice site in an engineered luciferase transgene containing an intronic insert with a T705G mutation[28]. Steric blockage of the aberrant splice site drives the use of a cryptic site which produces functional luciferase. The splice correcting PNA705 sequence makes an ideal cargo for this study because PNAs are not efficiently delivered to cells by classical CPPs. Further, once a delivery peptide is identified, PNA sequences can likely be changed to target other DNA/RNA sequences without significantly changing the physicochemical properties of the cargo or the efficiency with which it is delivered by a PNA delivery peptide (PDEP).

Two of the best known, classical CPPs are pTat$_{48–60}$ (tat) and pAntp$_{43–68}$ (penetratin). Although they can deliver some cargoes to cells under some conditions[29, 30], their ability to deliver PNA at low concentration is poor, as we show below. Here, we created a hybrid library from the aligned sequences of tat and penetratin. While each of the *parent* sequences is a CPP, their mechanisms of action differ, enabling the hybrid library to explore a broad

mechanistic space. At low concentrations (<10 μM), the cationic guanidinium-rich tat and its analogs, including nona-arginine (Arg9), enter cells mostly by endocytosis[31]. At higher concentrations, a mostly energy-independent mechanism of entry dominates as the peptide enters cells directly, perhaps after accumulation at ceramide-rich nucleation zones on the plasma membrane[8]. Penetratin is an amphipathic CPP that is capable of either direct translocation through the plasma membrane or translocation via the formation of a transient membrane structure[32].

In this work, the hybrid library was screened for PNA delivery efficiency, and PDEP daughter sequences are identified that deliver PNA with greatly improved efficiency at low concentration, and that significantly outperform both parent sequences. PDEPs conjugated to peptides, PNAs, PMOs, or other cargoes may represent powerful biotechnological tools. They may also comprise therapeutic delivery strategies that are fast and efficient, function at low micromolar concentrations in a variety of cell types, and have low cytotoxicity. More broadly, SME is shown here again to be a highly efficient approach toward the targeted optimization of peptide sequences.

## Results

**Library construction.** To evolve gain-of-function sequences from the known pTat$_{48–60}$ (tat) and pAntp$_{43–68}$ (penetratin) sequences, we created a peptide library of 8192 tat/penetratin hybrid sequences of 13–16 residues (Fig. 1). When aligned, the 13-residue tat sequence and 16-residue penetratin sequence share a lysine at position 4 and an arginine at position 10. We added a hydrophobic leucine option at position 10 to increase library diversity. Lys4 remains common to all sequences. This alignment creates a library with one cationic and one non-cationic residue possible at most positions. The three additional C-terminal residues of penetratin, Trp–Lys–Lys, were randomly present or absent as a cassette, resulting in 13 variable positions in peptides of 13 or 16 residues (Fig. 1c).

We used a split and recombine, one-bead one-sequence solid phase peptide synthesis strategy[33] to synthesize the CPP–PNA library on TentaGel Megabeads. C-terminal photo-labile linkers were coupled to each synthesis bead, followed by the 18-residue PNA705 sequence, followed by an AEAA PNA spacer, and then the N-terminal library member (Fig. 1a). High-pressure liquid chromatography (HPLC) and mass spectrometry on multiple individual beads showed that the synthesis was successful. After side-chain deprotection, beads were segregated into 96-well plates. PDEP–PNA705 constructs were cleaved from the resin with UV light, extracted into buffer, and incubated with HeLa *pLuc705* cells in serum-free media in 96-well plate format. Cells were incubated with peptide-PNA solution for 1 h and then in full media for 24 h before being analyzed by luciferase and bicinchoninic acid (BCA) assays. The initial PDEP–PNA705 concentration in each well was about 7.5 μM. After 24 h, cells were lysed and analyzed for functional luciferase and total protein. Positive PDEP sequences were defined as those that gave relative luminescence units (RLU) μg$^{-1}$ protein values that were at least three standard deviations above the plate mean (Fig. 1d). Beads from which positive sequences were extracted contained residual peptide that was sequenced by Edman degradation. Following a preliminary assessment of eight potential positives, we selected four, called P11, P14, P17, and P40, for subsequent detailed analysis (Fig. 1d).

**PDEP–PNA705 mediated splice correction.** To assess the efficiency of nuclear PNA705 delivery and splice correction, we quantified luciferase production at both the protein and mRNA

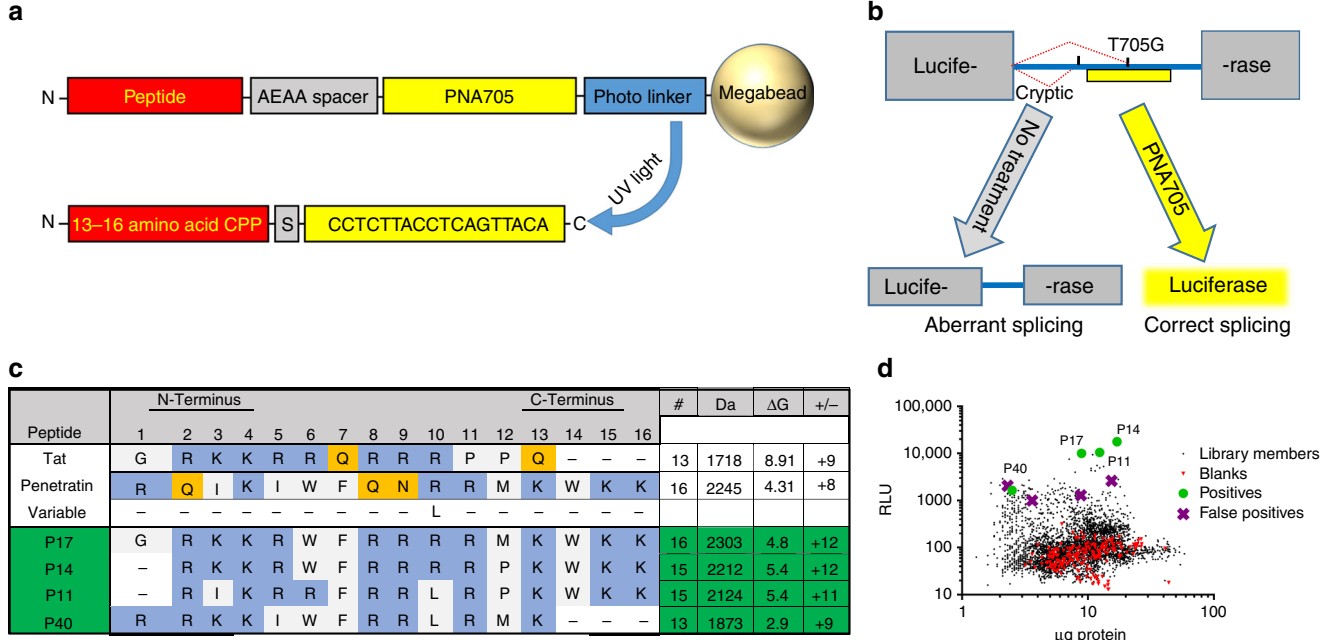

**Fig. 1** Selection of positive PNA delivery peptides (PDEPs). **a** Strategy for solid phase synthesis of photo-cleavable PDEP–PNA705 sequences onto TentaGel-S-NH₂ megabeads. **b** Screening system. HeLa pTRE-Luc IVS2-705 cells possess luciferase transgenes with mutant human β-globin IVS2 inserts that introduce an aberrant splice site at position 705 resulting in non-functional luciferase. Binding of PNA705 to the pre-mRNA in the nucleus blocks this splice site, enabling utilization of a cryptic splice site that restores functional luciferase production. **c** Split and combine synthesis outline. Each library bead contains ~0.5 nmol of one peptide sequence, with randomly determined residues either from the sequence of tat or penetratin. Residues 14–16 were randomly absent or present. At position 10, a hydrophobic leucine option was available. PDEP-positive sequences are in green. Basic amino acids are in blue and polar amino acids are in orange. **d** Plot of relative luminescent units vs. μg protein from the screen shows the distribution of library members, blank controls, PDEP-positive sequences, and four less active positive sequences. Peptides were photocleaved from the support resin, dissolved in ddH₂O, and added to cells in serum-free media. After 1 h, the cells were incubated in full media for 24 h before being analyzed by luciferase and BCA assays

level in HeLa *pLuc705* cells that had been treated with a 2× serial dilution of CPP-PNA705, starting at 5 μM. This concentration range is lower than the range at which CPPs are usually tested in vitro, providing a stringent test of delivery efficiency. PDEP–PNA conjugates are not toxic at this concentration range (Supplementary Figure 1). Cells were treated for 30 min in serum-free media at 37 °C, and then allowed to recover for 24 h in complete media. To assess the relative impact of endosomal uptake, which results in release to the cytosol, recycling, or lysosomal degradation, some cells were co-treated with the endosomal acidification inhibitor chloroquine (CQ) (120 μM), which promotes release over degradation. After 24 h, cell lysates were assayed for luciferase by luminescence (Fig. 2a) and for splice corrected mRNA by quantitative PCR (Fig. 2b). To control for total cell density, lysates were also analyzed for total protein by the BCA assay.

At 5 μM chimera, the parents do not efficiently deliver PNA705. Tat–PNA705 produced a slight 3-fold increase in luminescence over the background of untreated cells, while penetratin–PNA705 delivery produced a 15-fold increase over background (Fig. 2a). All four of the evolved PDEP daughter sequences (Fig. 1c) deliver PNA705 much more efficiently than the parent sequences, with P14 ≈ P11 > P17 > P40 >>pen > tat. The top-performing daughter sequences, P11 and P14, increased luciferase over background by more than 400-fold, which is more than 100 times better than tat and 30 times better than penetratin. Even at 1.25 μM, PDEP–PNA705 chimeras increase the splice corrected mRNA and functional protein over background by several fold. Interestingly, the canonical tat analog Arg9, which was not a member of the library, delivered PNA705 much better than tat itself. Yet, PDEPs P11, P14, and P17 outperformed Arg9 by 2–3 fold at 5 μM. CQ treatment moderately increased

luciferase for P11-PNA705, P17-PNA705, and P40-PNA705, while it had little effect on the delivery of P14-PNA705 suggesting moderate and sequence-specific levels of endosomal entrapment.

In addition to comparing PDEPs to the classical CPPs tat, penetrating and Arg9, we also tested a peptide called Peptide B which has recently been shown to drive efficient delivery of splice correcting PMOs in vivo[4, 34, 35]. PepB is (RXRRBR)₂ which has eight arginines and four linear linkers, X is β-alanine, and B is 6-amino-hexanoic acid. In our test system, peptide B has relatively low PNA delivery efficiency; higher than that of tat, but less than penetratin and Arg9 (Fig. 2a).

To quantify splice correction at the mRNA level, we designed qRT-PCR primers to target the splice junction of the properly spliced mRNA transcript. We measured the levels of corrected luciferase mRNA after 30-min incubations followed by a 24-h recovery period with 5 μM, 2.5 μM, and 1.25 μM PDEP-705 (Fig. 2d). These data validate the luciferase assay results. We observe a >100 fold increase in properly spliced luciferase mRNA in cells treated with PDEP–PNA705 relative to control. At 5 μM, the most efficient chimera, P14-PNA705, is 30-fold better than tat, 7-fold better than penetratin, and 4-fold better than Arg9. Even at 1.25 μM P14-PNA705, the increase in corrected mRNA is 2–3 fold, while tat and penetratin and Arg9 are not measurably higher than control.

**PDEP-TAMRA delivery to multiple cell types.** Parent and daughter sequences lacking the PNA cargo were characterized to determine the effect of the cargo on delivery. Sequences were labeled with the fluorophore tetramethylrhodamine (TAMRA) on a C-terminal cysteine residue (PDEP-TA) and their delivery to multiple human cell types was measured by flow cytometry.

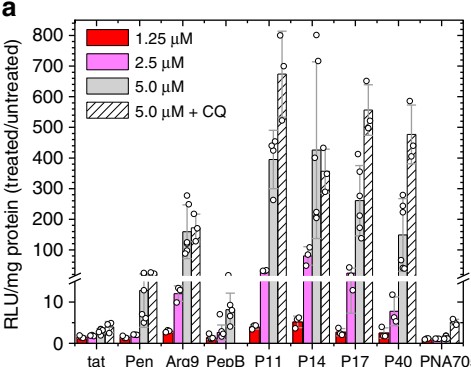
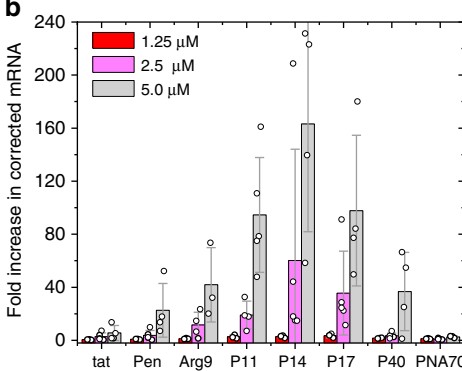

**Fig. 2** PDEP-mediated delivery of splice correcting PNA. **a** Luciferase expression. HeLa pTRE-LucIVS2-705 cells were incubated with varying concentrations of PDEP-705 in serum-free media for 30 min and then incubated for 24 h in complete media. Cell lysates were analyzed for luminescence and standardized by total protein content. The ratios of the RLU μg$^{-1}$ protein for treated samples divided by untreated samples are plotted. To examine the degree of endosomal entrapment, cells were co-incubated with 120 μM of the endosomolytic agent chloroquine. **b** mRNA correction. RNA was purified from lysates of cells treated as above and analyzed by qRT-PCR to determine the fold increase in correctly spliced luciferase mRNA standardized to ACTB1. The relative quantification of correctly spliced mRNA over untreated cells was calculated using the Pfaffl model[56]. Error bars are standard deviations of the data, which are shown as open circles, with 3–6 data points, from independent experiments, per average

To test the efficiency under stringent conditions, we used peptide concentrations of only 5 μM total containing 0.5 μM dye-labeled peptide. Peptides were incubated with HeLa, RAW264.7, HepG2, and MCF-7 cells for 30 min at 4 °C, 21 °C, or 37 °C, followed by trypsinization to detach cells for counting and to digest any externally bound peptide. PDEP-TA conjugates are not toxic at this concentration range (Supplementary Figure 2). Flow cytometric analysis (Fig. 3 and Supplementary Figure 4) was used to count TAMRA-positive cells and to measure TAMRA intensity distributions. Confocal microscopy was used to visualize peptide distribution in cells for each cell type (Fig. 3). Time series confocal images of PDEP P17-TA incubated HeLa cells at 21 °C and 37 °C (Fig. 3a) show immediate punctate membrane association. At 37 °C, but not at 21 °C, uptake of dye in punctate structures is observed. At both temperatures, individual cells rapidly and stochastically show the appearance of TAMRA with a diffuse cytosolic distribution in addition to bright puncta. All PDEPs behave similarly and maximum intensity is reached by 30 min at all temperatures. Cell surface-bound peptide is not observed under flow cytometry conditions (Fig. 3a).

PDEPs labeled with TAMRA cause diffuse cytosolic staining of many cells within 15 min of incubations at any temperature (Fig. 3), suggesting an energy independent mechanism of entry directly through the plasma membrane. This is consistent with the fact that similar delivery efficiency was measured at 37 °C, permissive for endocytosis, and at 4 °C and 21 °C, conditions under which endocytosis is inhibited. Cytotoxicity assays show no acute lactate dehydrogenase (LDH) leakage and no reduction metabolic potential or total protein in 24 h at concentrations ≤40 μM (Supplementary Figure 2). Inclusion of SYTOX Green in flow cytometry and confocal microscopy show that PDEP-TA conjugates enter cells without any membrane permeabilization.

While there was variability between cell types and incubation temperatures, daughter PDEPs consistently delivered much more TAMRA to a larger percentage of cells than either parent sequence or Arg9. For example, P17 delivered TAMRA to >98% of all four cell types, while tat and penetratin delivered TAMRA to less than 10% of cells in three of the four cell types. Furthermore, at all temperatures, the average intensity of intracellular TAMRA fluorescence is much higher for PDEPs across four different cell types than for the parent sequences (Fig. 3). For example, in HeLa cells at 21 °C, P14 and P17 deliver TAMRA to average intensities 10–20-fold higher than tat and

penetratin, and 4–6-fold higher than Arg9. In most experiments, P17 >>P11 = P14 > P40 = Arg9 > tat = penetratin which is similar to the efficiency of PNA705 delivery.

**PDEP delivery of GFP11 peptide**. To test the ability of PDEPs to deliver a protease-sensitive peptide cargo, we modified the well-known split-green fluorescent protein (GFP) assay[36] so that CPPs could be tested for the ability to deliver the polar 16-residue GFP11 peptide across the membrane barrier. The GFP11 peptide rescues the fluorescence of cytosolic, non-fluorescent GFP1–10 protein which is missing the GFP11 strand. Here, we stably transfected HeLa cells with a plasmid containing the full length mCherry fused to the non-fluorescent GFP1–10. Following incubation with PDEP-GFP11 constructs at multiple concentrations and temperatures for 30 min in serum-free media, the cells were analyzed by flow cytometry for mCherry and GFP fluorescence. Live single cells expressing mCherry (~50% of the total cells) were gated (Supplementary Figure 5) and their GFP fluorescence was analyzed (Fig. 4).

Unlike the luciferase splice correction assay where a single corrected mRNA transcript serves as template for many molecules of functional luciferase, the split GFP assay is stoichiometric; a single GFP11 peptide can bind to a single GFP1–10 protein producing a small, finite fluorescent response. Due to the decreased sensitivity for this system, we did not observe a measurable signal at 5 μM PDEP-GFP11, so we tested all peptides at 10–40 μM. Cytotoxicity assays show no LDH leakage, reduction in metabolic potential, or reduction in total protein at concentrations ≤160 μM except for P14D-GFP11, which showed a reduction in metabolic activity determined by alamarBlue assay at concentrations ≥20 μM despite also testing negative for cytotoxicity in either the LDH leakage or BCA protein assays (Supplementary Figure 3).

At 40 μM, the parent peptide tat delivers GFP11 to 21% of mCherry-positive cells at 37 °C, and to 7% of such cells at 21 °C (Fig. 4a). Penetratin does not measurably deliver GFP11 under any condition studied. The daughter PDEPs P11, P14, and P17 deliver GFP11 to more cells than the parents, at all concentrations, and at both temperatures. P14-GFP11 delivered the peptide to 59% and 46% of cells at 37 °C and 21 °C, respectively. Even at 20 and 10 μM, where both parent peptides are essentially inactive, PDEPs have a measurable GFP11 delivery capability (Fig. 4a).

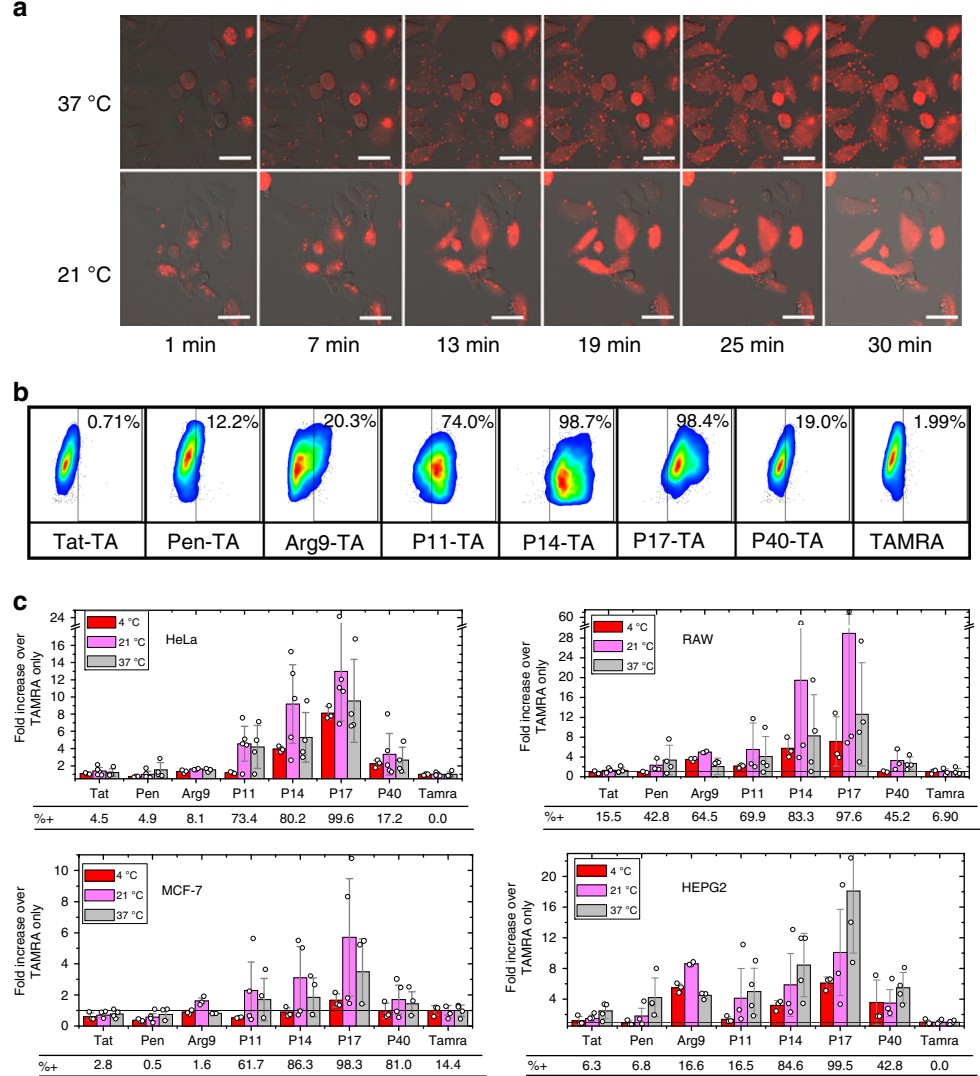

**Fig. 3** PDEP-mediated delivery of the fluorescent dye TAMRA. **a** Confocal microscopy images of HeLa cells incubated for 30-min with 4.5 μM P17-C + 0.5 μM P17-C-TA at either 37 °C or at room temperature (21 °C). The cytolysis-sensitive dye SYTOX Green was also added. Scale bar = 10 μM. **b** Example flow cytometry results for peptide-TA conjugates. Cells were trypsinized to remove externally bound peptides and subjected to flow cytometry. Cells were gated by scatter to select single viable cells. The very small fraction of SYTOX Green positive cells were excluded. The threshold for TAMRA positive intensity was set to be just above that of cells treated with free TAMRA dye, which does not enter cells. **c** For the four cell types shown, the averages of TAMRA fluorescence of live, SYTOX Green-negative cells were measured after treatment as above at 4 °C, 21 °C, and 37 °C. Averages are plotted as a ratio of the intensity of peptide-TA-treated cells to the intensity of cells treated with free TAMRA. Error bars are standard deviations of the data, which are shown as open circles, with 3–4 data points, from independent experiments, per average

In addition to delivering peptide to more cells, flow cytometric data show that the PDEPs deliver more peptide to each cell (Fig. 4b). The average GFP fluorescence in PDEP-GFP11 treated cells is 3–5-fold higher than background, while tat-GFP11 and pen-GFP11 are similar to background even at 40 μM. At 20 μM, P14-GFP11 and P17-GFP1 are up to 3-fold higher than background at 37 °C and at 21 °C. At 10 μM, where most conjugates exhibit background fluorescence levels, P14-GFP11 remains 44% above background at 37 °C and 72% above background at room temperature. P14 is most efficient at GFP11 delivery, followed by P11 and P17. Interestingly, like penetratin, the PDEP P40 does not deliver GFP11 significantly, although it successfully delivers PNA705 (Fig. 2).

To test the potential effect of CPP proteolysis on delivery efficiency, we included in this assay D-amino acid versions of both tat and P14, coupled to L-amino acid GFP11. In both cases, the D-form CPPs outperformed the L-form CPPs by a moderate amount, indicating that proteolytic sensitivity has some effect on delivery efficiency, although not a dominant one. At 40 μM, tat (D)-GFP11 was delivered to more cells and at a higher concentration that tat-GFP11 at both 37 °C and 21 °C. GFP fluorescence just above background was observed at lower concentrations for tat(D)-GFP11 but not for tat-GFP11. Similarly, P14(D)-GFP11 outperformed P14-GFP11 under all other conditions (Fig. 4). In fact, P14(D)-GFP11 was the only peptide tested that had measurable activity at 10 μM, with a >2-fold fluorescence increase, over background at both temperatures.

## Discussion

The global market for transfection reagents and equipment is forecast to reach $1.02B in 2021, up from $715.4M in 2016 according to Markets Research global forecast report[37]. Interest in lipidic and peptidic delivery strategies are expected to grow as

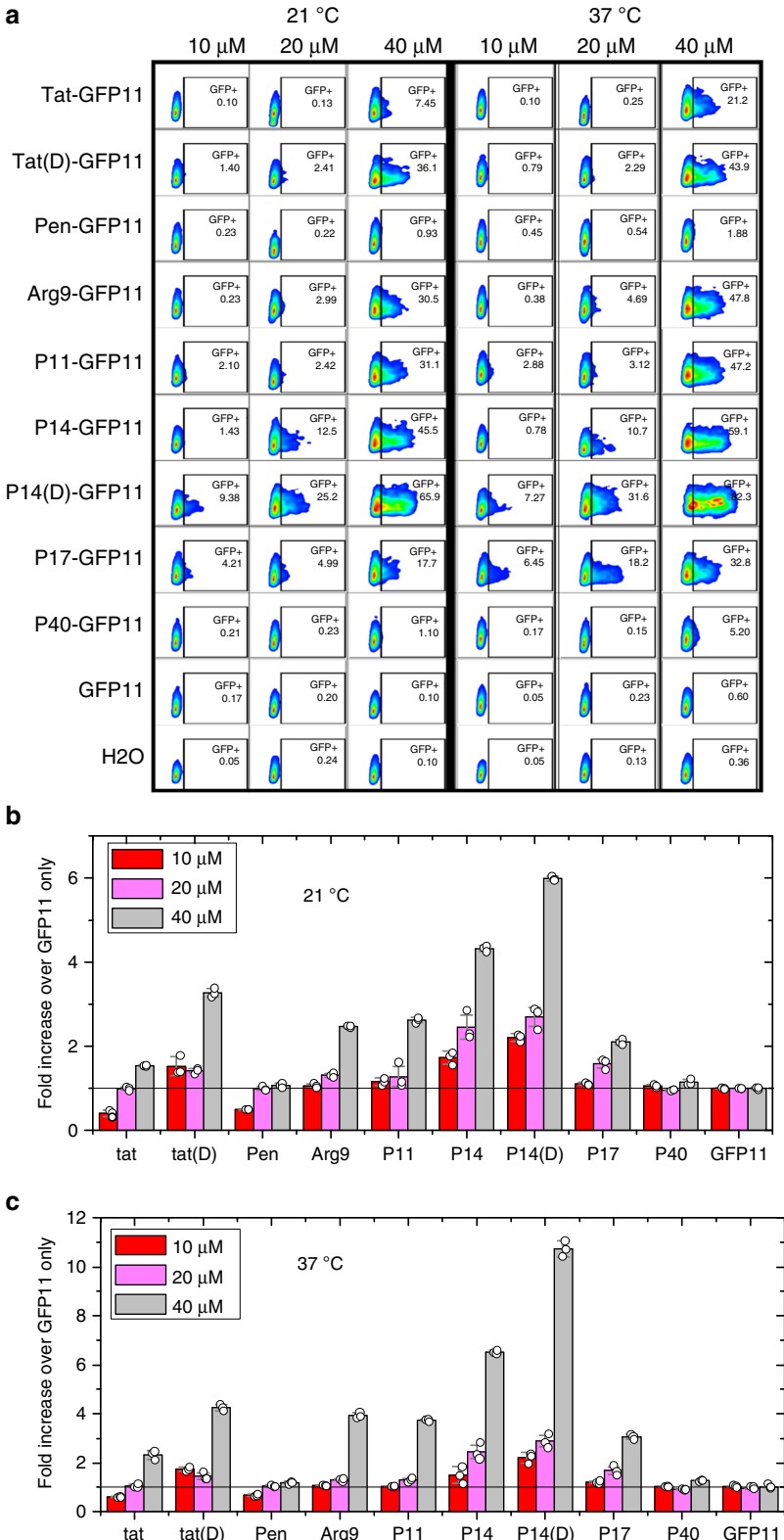

**Fig. 4** PDEP-mediated delivery of an anionic peptide cargo. **a** Cells were transfected with a plasmid encoding for intact mCherry and non-fluorescent GFP1–10. Cells were incubated, either at room temperature or at 37 °C, with varying concentrations of Peptide-GFP11 for 30 min and analyzed by flow cytometry. Live, single, mCherry-positive cells were first gated. Within this population, mCherry and GFP intensities were measured for about 10,000 cells for each sample. The percentage of GFP-positive cells shown was determined by gating just above background GFP levels. **b**, **c** Average GFP intensity for cells treated at 21 °C (**b**) and at 37 °C (**c**) with CPP-GFP11 constructs. Averages are divided by the average intensity of cells treated with the GFP11 peptide alone. D-form versions of tat and P14 conjugated to an L-form GFP11 peptide were also analyzed. Error bars are standard deviations of the normalized data, which are shown as open circles, with three data points, from independent experiments, per average

efficiencies improve and costs decrease. Similarly, the peptide therapeutics market was valued at $17.5B in 2015 and is expected to increase at a compound annual growth rate of ~10% through 2025 as researchers and drug companies look to bypass impediments that arise with small-molecule therapeutics from off-target interactions, low response rates, and drug resistance[2, 38]. While antisense oligomers have not yet reached the therapeutic marketplace, the potential for a substantial market is clear[3, 39]. The development of peptide, PNA, PMO, or other polymer drugs targeting intracellular ligands requires that they efficiently enter cells. While peptide and polymer conjugation, micro-reservoir delivery systems, and cell type-specific targeting moieties exist and address some pharmacokinetic impediments to such drugs[40], effective strategies and new modalities for efficient intracellular delivery of macromolecules are still needed to advance the fields.

CPPs as biotechnological tools[41] and in therapeutic delivery[42] constitute a valuable and growing strategy toward solving the problem of intracellular delivery of membrane impermeant cargoes. There are examples of successful delivery of some cargo types, especially fluorescent dyes that are covalently attached to CPPs[9], and oligonucleotide cargoes that are non-covalently complexed with CPPs[41]. However, delivery of other classes of cargoes, including proteins, peptides, PNA, PMO, and others will benefit from improvements in CPP efficiency in vitro and in vivo. Further, the field will benefit from new processes for rapidly improving upon known CPPs, building on some successes in screening and algorithm-aided design[43, 44].

The emerging consensus on the efficiency and mechanism of action of classical CPPs such as tat and penetratin[9, 45, 46] is that direct translocation and endocytosis-dependent uptake and escape both occur in parallel, but at rates that depend on CPP sequence and concentration, cell type, cargo properties, buffer, temperature, and other experimental variables. Efficient delivery of cargo with these classical CPPs in vitro often requires high concentration that leads to direct translocation across the membrane, but simultaneously can also cause membrane disruption and toxicity[47]. The lack of simple sequence–function relationship rules has impeded the engineering and design of CPPs with higher efficiencies, or with altered reliance on specific mechanisms of entry. With a few exceptions, discovery of new CPPs is driven by trial and error experimentation.

In the absence of explicit sequence–structure–function relationships to guide rational improvements, we reasoned here that SME would be a powerful strategy for systematically discovering more efficient CPPs. We thus synthetically evolved a family of peptides for efficient delivery of an 18-residue PNA to cells. PNA is a class of non-natural antisense oligomer that binds with high sequence specificity to DNA and RNA and can modulate transcription, translation, splicing, and replication[20–27] and can have such activity in vivo[26, 27]. While these properties make PNA an exciting candidate biotechnological tool and therapeutic, efficient and routine intracellular delivery remains a challenge.

We simultaneously optimized for efficient delivery of a polar PNA cargo of 5 kDa, low toxicity, and solubility of the CPP-PNA chimera using a hybrid library. The daughter PDEPs identified in the screen (Fig. 1d) are more than two orders of magnitude more active than the parent sequences at 5 μM, despite the fact that all daughter sequences are 50–60% identical to the parents. The daughters are more cationic than the parent CPPs, with charges ranging from +9 to +12. The polar residues Asn and Gln, possible in five different positions in the library, are completely excluded from the positive sequences in favor of basic residues Lys and Arg, or the non-polar Phe. The two top-performing sequences selected from the library only differ at Pos12, Pro/Met, and Pos1-/Gly. In both sequences, Trp was selected over Arg at Pos 6 and the Trp-containing terminal cassette at Pos14–16 is

present. The third most active PDEP, P11, has only the Trp containing terminal cassette with an Arg at Pos6, but is otherwise similar to P14. P40 consistently outperforms the parent sequences but underperforms the other PDEP positives. It contains the Trp at Pos6 but lacks the Trp containing cassette at Pos14–16. None of the less active positive sequences rejected for further testing (Fig. 1) contained a Trp in either position 6 or the cassette. In synthetic vesicles, the tryptophans in penetratin have been shown to insert themselves into the membrane and their degree of evolutionary conservation among homeoproteins suggests functional importance[48, 49]. A relationship likely exists between hydrophobicity and amphipathicity, driven by the number and location of Trp in the peptide sequence, with increasing hydrophobicity being associated with more efficient membrane translocation and amphipathicity with increasing cytotoxicity[50]. Taken together, these observations suggest the possibility for further fine-tuning of the cargo-delivery capabilities of CPPs by allowing the location and number of Trp residues in the sequence to vary in a new round of molecular evolution.

The PNA705 model ultimately requires nuclear delivery of the 18-residue PNA to target pre-splice luciferase mRNA (Fig. 1b). Cytosolic delivery is achieved via a combination of direct plasma membrane translocation and endocytosis and escape, while nuclear delivery from the cytosol is likely achieved via passive diffusion through the nuclear pore, and perhaps some nuclear localization signal-like activity encoded in the cationic CPP sequence. While the magnitude of the PDEP improvement over the commonly used tat and penetratin parent sequences observed at 5 μM is dramatic, perhaps even more promising is that we observe increases in corrected luciferase mRNA for 1.25 μM PDEP–PNA705 (Fig. 2d), a concentration at which few CPPs, including the parent sequences, deliver any cargo measurably. While this level of response means incorrectly spliced mRNA is still dominant, in many diseases, including cystic fibrosis, hemophilia B, F7 deficiency, Fanconi anemia, Bardet-Biedl syndrome, propionic acidemia, retinitis pigmentosa, and Duchenne muscular dystrophy, the production of even a small amount of functional protein has the potential to dramatically improve the phenotype[3].

We anticipate that the PDEP–PNA constructs developed in this work may help to catalyze, for PNA, additional progress like that made in the application of CPP–PMO chimeras toward the correction of splice defects in vivo[4, 39, 51, 52]. In the PMO field, development of some CPPs, mostly by trial and error, has resulted in CPP–PMO chimeras that correct splice defects in small animal models of diseases such as Duchenne muscular dystrophy[35, 53–55].

To assess the degree to which the improved performance of the PDEP daughter sequences over the parent sequences, and the mechanism of entry, was specific to the PNA cargo used in the screen, we tested delivery of peptides labeled with the dye TAMRA and peptides conjugated to the anionic GFP11 peptide. Daughter PDEPs effectively deliver GFP11 and TAMRA to multiple cell types at multiple temperatures with efficiencies that are much higher than the parent peptides. However, the rank order is not the same for all cargoes. For example, P17 is the most efficient PDEP for TAMRA delivery, but is outperformed by P11 and P14 for PNA705 and GFP11 delivery. P40 delivers PNA705 efficiently, but does not measurably deliver GFP11. These observations of some cargo dependence on delivery efficiency, even among very similar CPPs is an argument for screening for delivery using the intended cargo, or a close physical chemical analog, rather than a surrogate cargo such as a fluorescent dye.

PDEP enter cells by multiple mechanisms simultaneously. Direct plasma membrane translocation of PDEPs with cargo is evidenced by efficient delivery at 21 °C (room temperature) and at 4 °C, where energy-dependent endocytosis is slow or absent, and

is supported by confocal microscopy measurements showing rapid appearance of diffuse cargo directly in the cytosol after peptide addition. Endocytosis-dependent uptake of CPP-cargo molecules is evidenced by some punctate intracellular fluorescence and by moderately increased delivery at 37 °C compared to lower temperatures. This is presumably observed because the relative contribution of endocytosis is increased at 37 °C, but perhaps also because the membrane physical properties become more permissive for translocation at 37 °C. Some contribution of endosomal uptake is also supported by the fact that the degree of PNA705 delivery is moderately enhanced, for some, by co-treatment with the endosomolytic agent CQ. Finally, D-amino acid *tat* and PDEP P14 both delivered protease-sensitive, L-amino acid GFP11 with somewhat increased efficiency, suggesting a role for endosomal, lysosomal, or cytosolic proteases.

The SME screen in this study was specifically designed to identify sequences capable of delivering PNA cargo, although the sequences identified proved capable of delivering other cargoes. SME generates the most active sequences from the queried sequence space for satisfying the specific requirements of the screen. The power of the technique is derived from the ability to design a screen to best select the desired traits and from the ability of the library to reflect current knowledge such that variations in the library can represent testable hypotheses about the mechanism of action. By identifying optimal sequences to overcome specific delivery challenges, SME has the potential to redefine the way CPPs are generated, optimized, and used in both the laboratory and the clinic moving forward.

As macromolecular tools and therapeutics become more prevalent and the demand for intracellular delivery of macromolecules increases, the need for efficient delivery strategies is becoming more pressing. Natural selection has produced CPP sequences, such as *tat* and penetratin, which function to deliver particular cargoes to cells. Yet, they are often not efficient enough to be useful as generic delivery vehicles in the lab or clinic. Here, we have used SME to identify gain-of-function CPPs with dramatically improved ability to deliver PNA, peptide, and other cargoes to cells in culture. Thus, SME is shown to be a powerful technique for rapidly improving upon existing functional peptide sequences, not only in the field of CPPs and delivery, but in all fields in which bioactive peptides are studied. Specifically, the insights gained in this study will inform the design of iterative libraries to generate PDEPs with additional desirable traits such as cell and tissue targeting, resistance to serum inhibition, increased circulation, and more as we continue to work toward the dual goals of developing (i) useful tools for the biotechnology lab, and (ii) useful delivery vehicles for systemic therapeutics.

## Methods

**PDEP–PNA synthesis**. PDEP–PNA library synthesis was performed using a split and combine synthesis strategy on TentaGel MB NH₂ resin (MB300_002) loaded with Fmoc-photolabile linker (Advanced ChemTech RT1095) using standard Solid Phase Peptide Synthesis (SPPS) protocols. FMOC protected peptide (Advanced ChemTech)/PNA monomers (PNABio FMA001, FMT001, FMG001, FMC001, FMO001) were dissolved in DMF at 3× molar excess relative to the manufacturer's stated loading capacity. The reaction was catalyzed by the addition of 0.9× molar HBTU/HOBt or 0.9× molar HATU for peptides and PNAs respectively. All bases were double coupled (2×20-min reactions) and reaction completion was demonstrated with the Kaiser test for amines. Following the addition of each base, remaining reactive sites were capped with 50× molar acetic anhydride and DIPEA. Between the peptide and the PNA, two [2-(2-(Fmoc-amino) ethoxy) ethoxy] acetic acid spacer moieties were added. Following acid deprotection in Reagent B (88% v/v trifluoroacetic acid, 5% v/v phenol, 5% v/v ddH₂O, 2% v/v triisopropylsilane) supplemented with 2.5% v/v m-cresol, beads were washed thoroughly in dimethylformamide and dichloromethane and dry-cleaved from the solid support under UV light for 4 h to produce amidated sequences. Synthesis quality was verified by performing HPLC, mass spectrometry, and Edman sequencing on the peptide released from individual beads. Individual beads were placed in 96-well plates, suspended in 200 μl hexafluoro-2-propanol, and placed under UV light at 365 nm

until completely dry. Cleaved peptide–PNA sequences were dissolved in 50 μl ddH₂O and were added to cells that were used in subsequent luciferase/BCA assays.

Peptide-PNA705 conjugates were synthesized as described above using Tentagel XV-NH₂ resin (Rapp Polymere XV18130.002) starting from the C-terminal residue of the PNA. Conjugates were acid cleaved, ether precipitated, dissolved in glacial acetic acid and lyophilized. Purity was assessed on HPLC and mass was verified by MALDI mass spectrometry.

**TAMRA labeling**. Amidated peptide sequences with a C-terminal cysteine were ordered from Biosynthesis (www.biosyn.com) and dissolved at 2 mg ml⁻¹ in degassed PBS at pH 7. A 100× molar excess of tris-carboxyethylphosphine (TCEP) was added to ensure complete reduction of the C-terminal thiol group. Tetramethylrhodamine-5-Maleimide was dissolved in degassed dimethylformamide (DMF) and added to the peptide solution at 10× molar excess. The reaction mixture was topped with nitrogen gas and stored overnight at 4 °C. Excess dye was removed using cationic exchange resin (PolyLC TT1000CAT). Labeled peptide was eluted in 25% glacial acetic acid in ddH2O and lyophilized. Purity and mass were verified by HPLC and MALDI-TOF. Concentrations were determined by absorbance at 556 nm using $\mathcal{E} = 89$ L mmol⁻¹ cm⁻¹.

**High-pressure liquid chromatography**. All PNA and peptide–PNA constructs were analyzed and purified by reversed-phase chromatography. The stationary phase was a 100 mm × 4.6 mm C-18 column from Kromasil. The mobile phase was composed of a gradient of distilled water (0.1% trifluoroacetic acid) and acetonitrile (0.1% trifluoroacetic acid) with a flow rate of 1 ml min⁻¹. Where possible, peptide and peptide fragments were analyzed using tryptophan fluorescence (285ex/340em) and PNA-containing compounds were analyzed using absorbance at 260 nm. In the absence of tryptophan or PNA residues, peptide was analyzed by absorbance at 220 nm.

**MALDI mass spectrometry**. PNA and peptide–PNA constructs were mass verified using a Bruker Autoflex III MALDI–TOF mass spectrometer (Bruker Daltonics). Mass spectra data were collected in both linear and reflector mode with positive ion detection. Typical sample preparation for MALDI–TOF data was performed by making stock solutions of 70% acetonitrile: H₂O + 30% H₂O with 0.1% trifluoroacetic acid saturated with α-Cyano-4-hydroxycinnamic acid matrix (20 mg ml⁻¹). Ten microliters of the stock solution was mixed with 1 μl PNA/peptide–PNA solution at 10–1000 μM, deposited onto the MALDI target plate and allowed to evaporate via the dried droplet method.

**Generation of HeLa p_mCherry-GFP1–10 cells**. HeLa cells were transfected with 2500 ng of linearized (BamH1) mCherry-GFP1–10 (Addgene Plasmid #78591) using Lipofectamine 3000 following standard protocols and maintained under G418 selection at 900 μg ml⁻¹; 10⁶ cells were analyzed by flow cytometry using a BD FACSARIA III and individual cells expressing high mCherry concentrations were sorted one cell per well into 96-well plates. Stable clones were selected, expanded, and frozen.

**Cell culture**. HeLa pTRE-LucIVS2-705 cells in this study were generously donated by Dr. Rudolf Juliano (University of North Carolina, Chapel Hill). HeLa pTRE-LucIVS2-705, HeLa p_mCherry-GFP1–10, HeLa, RAW264.7, HepG2, and MCF-7 cells were obtained from ATCC. Cells were cultured at 37 °C with 5% CO₂ in Dulbecco's Modified Eagle's Medium (DMEM) (Gibco) supplemented with 10% fetal bovine serum (FBS) (Gibco), 1% antibiotic–antimycotic (Gibco), and 1% non-essential amino acids (Gibco). Cells were passaged 1:5 at 90% confluency. All assays were conducted on cells passaged fewer than 10×.

**PNA705 mediated luciferase splice correction**. HeLa pTRE-LucIVS2-705 cells were seeded on flat-bottom 96-well plates at 10,000 cells per well in 200 μl complete DMEM. The following day, cells were washed in phosphate buffered saline and topped with premixed 50 μl 2× serum/phenol red-free DMEM (Gibco) + 50 μl peptide cleavage solution (ddH₂O + peptide). Cells were transfected for 30 min at 37 °C with 5% CO₂. The transfection solution was removed, cells were topped with 100 μl complete DMEM. The next day (24-h recovery) cells were washed in PBS. Cells were lysed in 20 μl reporter lysis buffer (Promega E4030) following the manufacturer's protocol. A volume of 10 μl of lysate from each well was transferred to a 96-well solid black plate; 100 μl of luciferase assay reagent from the Luciferase Assay System (Promega E1500) was added, the plate was agitated for 10 s, and luminescence was measured with a BioTek Synergy 2 plate reader with injector ports. Total protein was measured with a BCA assay (described below). Data were expressed as RLU per μg protein (RLU (μg protein)⁻¹). During screening, positive wells were defined as those whose RLU per μg protein values exceeded the average for the plate by at least three standard deviations. Subsequent PNA705 delivery experiments were conducted as described above except that volumes were scaled for 24-well plate format.

**BCA assay**. The remaining 10 μl of lysate from the splice correction assay was combined with 100 μl of the BCA reagents and incubated for 1–4 h at 37 °C per the

manufacturer's protocol. Absorbance at 562 nm was measured using a measured with a BioTek Synergy 2 plate reader.

**LDH assay**. HeLa cells were treated for 30 min at 37 °C with 100 μl of a 2× serial dilution of PDEP constructs ranging from 160 μM to 1.25 μM in serum-free DMEM. As a positive control for LDH release, cells were treated with lysis buffer. Cells incubated in serum-free DMEM lacking peptide were used as a negative control, 50 μl of the incubation solution was transferred to a separate 96-well plate, 50 μl of the reaction mixture was added to each well and incubated for 30 min at room temperature in the dark, and 50 μl of stop solution was added to each well and absorbance was read at 490 and 680 nm. % cytotoxicity was calculated as follows:

$$[(LDH@490\,nm) - (LDH@680\,nm)] = LDH\ Activity, \tag{1}$$

$$\%Cytotoxicity$$
$$= \frac{(PDEP - Control\ treated\ LDH\ Activity) - (Spontaneous\ LDH\ Activity)}{(Maximum\ LDH\ Activity) - (Spontaneous\ LDH\ Activity)} \times 100. \tag{2}$$

**AlamarBlue assay**. A total of 24 h after PDEP treatment, HeLa cells in 100 μl media were treated with 10 μl of 10× alamarBlue reagent and incubated for 2 h at 37 °C. Fluorescence at ex570/em585 was measured and standardized to untreated wells.

**qRT-PCR**. HeLa pTRE-LucIVS2 cells were treated as described above with PDEP–PNA705 constructs at varying concentrations. RNA was extracted using the Direct-Zol™ RNA purification system (Zymo Research) and reverse transcribed with the iScript cDNA synthesis kit (Bio-Rad), and mRNA splice correction was detected using PowerUp SYBR Green master mix (ThermoFisher) on an Applied Biosystems QuantStudio 6 using the following primers sets:
ACTB1: 5′-CCTTGCACATGCCGGAG-3′
ACTB2: 5′-ACAGAGCCTCGCCTTTG-3′
pTRE-Luc IVS2 Downstream: 5′- TCAATCAGAGTGCTTTTGGCG-3′
pTRE-Luc IVS2-Bridge: 5′-TTACGATCCCTTCAGGATTACAA-3′
The relative quantification of correctly spliced mRNA (90 bp amplicon) over background was calculated using the Pfaffl model[56].

**TAMRA delivery assay**. HeLa, RAW264.7, HCT116, HepG2, and MCF-7 cells were plated in 12-well plates at 100,000 cells/well. After 24 h, cells were gently washed in PBS, topped with 500 μl of PDEP-TA solution containing 0.5 μM PDEP-TA + 4.5 μM PDEP-C in sfDMEM without phenol red, and incubated for 30 min at either 4 °C, 23 °C, or 37 °C. The incubation solution was aspirated and cells were treated with 100 μl 0.025% Trypsin for 3 min at 23 °C; 500 μl DMEM containing 2% FBS, 125 nM Sytox Green, and 20 mM HEPES was used to suspend the cells. Cells were transferred to a filter-topped flow cytometry tube and analyzed on a BD LSR II flow cytometer. Cells displaying a normal morphology were gated and first analyzed for GFP fluorescence using the 488 nM laser. GFP-negative cells were used to generate a histogram of TAMRA fluorescence, measured with the 543 nM laser. The calculated mean of this distribution and the percentage of cells exhibiting TAMRA fluorescence values higher than $10^3$ were recorded.

**Confocal microscopy**. Cells were treated as described above for the TAMRA delivery assay and immediately imaged to avoid artifacts resulting from fixation. The distribution of TAMRA and Sytox Green was analyzed using a confocal scanning Nikon Eclipse Ti2 inverted microscope using a 40× oil-immersion objective. Sytox Green was excited using the 488 nM laser and TAMRA was excited using the 543 nM laser.

**Split GFP assay**. HeLa cells were transfected with p_mCherry-GFP1–10(29) (Addgene plasmid #78591) using Lipofectamine 3000 (ThermoFisher). Single mCherry-positive cells were sorted into 96-well plates using a BD FacsAria cell sorter and colonies were maintained under G418 selection at 600 nM. Stably transfected cells were plated in 12-well plates at 100,000 cells/well and grown overnight in complete DMEM. Cells were washed in PBS and incubated in 500 μl of PDEP-GFP11 solution at variable concentrations at 23 °C and 37 °C for 30 min. The PDEP-GFP11 solution was aspirated and 100 μl of 0.025% trypsin solution was added to each well. After detachment, cells were suspended in 500 μl of PBS + 2% FBS and 20 mM HEPES. Cells were analyzed using a BD LSRII flow cytometer. Cells displaying a normal morphology were gated and first analyzed for mCherry fluorescence using the 543 nM laser. mCherry+ cells were used to generate a histogram of GFP fluorescence measured with the 488 nM laser. The calculated mean of this distribution and the percentage of cells exhibiting GFP fluorescence values higher than $10^3$ were recorded.

**Data availability**. All data are available from the corresponding author upon request.

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

## Acknowledgements

The authors wish to thank Rudolf Juliano at the University of North Carolina for kindly providing HeLa *pLuc705* cells. We also thank Scott Grayson for assistance with MALDI mass spectrometry, and Dorota Wyczechowska at the Louisiana State University core facility for assistance with flow cytometry. Funded by NIH NIGMS R01GM111824 and by the Louisiana Board of Regents Support Fund.

## Author contributions

W.B.K. designed and performed most of the experiments. S.G. designed and performed the experiments. W.B.K., S.G., and W.C.W. analyzed the data. W.B.K. drafted the manuscript which was edited and revised with W.C.W. and S.G.

## Additional information

**Competing interests:** W.B.K. and W.C.W., together with Tulane University, have applied for a patent based on the intellectual property contained herein. The remaining authors declare no competing interests.

