## [Peer Review File · Nature Communications]

Reviewers' comments:

Reviewer #1 (Remarks to the Author):

The method of synthetic molecular evolution (SME) was applied to identify CPPs with improved ability to deliver PNA to cell nuclei. A CPP library containing 8,192 tat/penetratin hybrid peptides was screened using the HeLa pTRE-LucIVS2 (HeLa705) splice correction reporter system. The best PNA Delivery Peptide (PDEP) daughter sequence drove 180-fold and 30-fold better luciferase expression than the parents tat and penetratin, respectively. The daughter PDEPs also deliver a 19-residue peptide cargo to cells much more efficiently than either parent sequence, and deliver a dye cargo with high efficiency into multiple cell types.

In this work, a dramatic improvement of CPPs in performance following a single iteration of SME demonstrates the power of this approach to peptide sequence optimization. The results are exciting and the novel CPPs are a valuable research tool, and a possible therapeutic vehicle, for delivery of membrane impermeable PNA or peptide sequences.

This work is ready for publication

Reviewer #2 (Remarks to the Author):

Within this current paper the authors describe the creation of a peptide library for identifying novel hybrid cell-penetrating peptides (CPPs). CPPs are widely used as tools to enhance the intracellular delivery of various molecules. The peptides reported herein comprise sequences from the well-studied CPPs tat and penetratin. A huge peptide library has been designed (>8000) and members were explored particularly for their ability to transduce peptide nucleic acids. Moreover, the authors gained deeper insights into the activity of some of the most effective new peptides by quantifying the cellular uptake and cargo delivery ability.

The method the authors applied, synthetic molecular evolution, to identify promising new CPPs is not new, but seems to be very powerful to identify suitable candidates. The paper is concise and well-written, the data seems to be valid and reproducible. The topic of the paper is of high interest, since methods for efficient drug delivery are still of high need. Although at some points very detailed, the paper might be also interesting for people not directly working with CPPs but that are interested to do so in the future.

I have only some minor comments to this work:

1. Figure 1A: The sketch of the peptide should be drawn from left (C-terminal) to right (N-terminal), as is the convention for showing peptide sequences...
2. Page 5, first paragraph: Only 4 hit peptides were identified/selected. Can the authors comment if this number is high or low. For me, it seems to be a relatively low outcome after screening a library of >8000 peptides?
3. Page 5, second paragraph: The authors state that they did not observe any long-term effect on cell viability. What do they mean with "long-term"? Please specify.
4. Figure 3: SD seem to be relatively high, especially for conditions 21/37°C. Why is this the case? This observation should be discussed in the text. Did you test higher concentrations? May be this is the reason. Later on the authors use concentrations >10 µM at least.
5. Page 15: The authors should also include other library screens on CPPs within their discussion (e.g. Lin et al. *Angewandte Chem Int Ed* 2015, 54, 10370).

Reviewer #3 (Remarks to the Author):

The authors have a solid experience in synthetic molecular evolution (SME) of peptides. They now apply this interesting strategy to the synthesis of CPP-PNA libraries which has never been done to

our knowledge.

Several publications indeed demonstrated that the Tat and Penetratin CPPs were poorly efficient to promote the delivery of antisense oligonucleotides analogues (such as PNA or PMO) geared at pre-mRNA splicing redirection. This has been demonstrated in the easy-to-implement HeLa705 splice correction assay and more recently in DMD (Duchenne muscular dystrophy) cells in vitro and in vivo using mdx mice.

In their short review of the literature, the authors seem to ignore that recent studies (by the groups of H.Moulton, M.Gait and others) have led to the discovery of new CPPs with a much higher delivery potential of conjugated PNAs (or more frequently PMOs) using these assays. Discussing these CPPs and comparing their efficiencies with the author's PDEPs should be done at least in the HeLa 705 assay. Comparing dose-response curves for selected PDEPs and control CPPs would help appreciating the claimed superiority of PDEPs. Comparison with oligoArg-based constructs indicates a modest (around 3x) increase in efficiency.

Supplementary material does show higher toxicity of the PDEP-PNA conjugates as opposed to the parent Tat and Antp CPPs. This could be due to the increased Arg content of PDEPs and might become a problem for further clinical implementation. Such problems are considered as serious ones for the potential use of CPPs in DMD treatment.

The chloroquine (CQ) experiments should in principle be indicative of entrapment in endocytic vesicles. Somewhat puzzling however is the fact that CQ does not have any effect on oligoArg, Tat or Pen based conjugates. Fluorescence microscopy experiments with fluorochrome-tagged PNAs should help defining the efficiency with which PNAs are delivered to cell nuclei.

At variance with most recent data in the literature for several CPPs, the uptake of PDEP-TAMRA conjugates is poorly affected by low temperature incubation in line with a direct translocation across the plasma membrane. This is somewhat in contradiction with CQ data and with the kinetics of cell uptake. Studies of cell trafficking should be done with fluorochrome-tagged CPP-ON conjugates and not with labelled CPPs as done here. It is well known indeed that the CPP cargo influences the overall behavior of the constructs. Accordingly, FACS and confocal microscopy experiments with fluorochrome-tagged PDEP-PNA should be done and would help discriminating between direct membrane translocation and endocytosis.

In the last part, the authors investigated the potential of their peptides to deliver a 16-mer GFP1 peptide using the split-GFP assay. Again increased efficiency was achieved with PDEPs as compared to Tat, Pen or Arg although very high concentrations had to be used.

Additional experiments and a clear demonstration of increased efficiency as compared with recently described CPPs needs to be done before acceptance.

Synthetic molecular evolution of hybrid cell penetrating peptides.

Authors: W. Berkeley Kauffman, Shantanu Guha, and William C. Wimley

NCOMMS-17-33271A

Reviewers' comments:

Reviewer #1 (Remarks to the Author):

Comment: The method of synthetic molecular evolution (SME) was applied to identify CPPs with improved ability to deliver PNA to cell nuclei. A CPP library containing 8,192 tat/penetratin hybrid peptides was screened using the Hela pTRE-LucIVS2 (Hela705) splice correction reporter system. The best PNA Delivery Peptide (PDEP) daughter sequence drove 180-fold and 30-fold better luciferase expression than the parents tat and penetratin, respectively. The daughter PDEPs also deliver a 19-residue peptide cargo to cells much more efficiently than either parent sequence, and deliver a dye cargo with high efficiency into multiple cell types.

In this work, a dramatic improvement of CPPs in performance following a single iteration of SME demonstrates the power of this approach to peptide sequence optimization. The results are exciting and the novel CPPs are a valuable research tool, and a possible therapeutic vehicle, for delivery of membrane impermeable PNA or peptide sequences.

This work is ready for publication

OUR RESPONSE: We appreciate the reviewer's positive comments.

Reviewer #2 (Remarks to the Author):

Comment: Within this current paper the authors describe the creation of a peptide library for identifying novel hybrid cell-penetrating peptides (CPPs). CPPs are widely used as tools to enhance the intracellular delivery of various molecules. The peptides reported herein comprise sequences from the well-studied CPPs tat and penetratin. A huge peptide library has been designed (>8000) and members were explored particularly for their ability to transduce peptide nucleic acids. Moreover, the authors gained deeper insights into the activity of some of the most effective new peptides by quantifying the cellular uptake and cargo delivery ability.

Comment: The method the authors applied, synthetic molecular evolution, to identify promising new CPPs is not new, but seems to be very powerful to identify suitable candidates. The paper is concise and well-written, the data seems to be valid and reproducible. The topic of the paper is of high interest, since methods for efficient drug delivery are still of high need. Although at some

points very detailed, the paper might be also interesting for people not directly working with CPPs but that are interested to do so in the future.

I have only some minor comments to this work:

Comment: 1. Figure 1A: The sketch of the peptide should be drawn from left (C-terminal) to right (N-terminal), as is the convention for showing peptide sequences...

OUR RESPONSE: In Figure 1, we have flipped the images of the synthesized chimeras so that the direction matches the convention of N-terminus on the left. We have also added the sequence of the PNA to that image and adjusted the image of the released molecule to reflect the relative sizes of the parts.

Comment: 2. Page 5, first paragraph: Only 4 hit peptides were identified/selected. Can the authors comment if this number is high or low. For me, it seems to be a relatively low outcome after screening a library of >8000 peptides?

OUR RESPONSE: As can be seen in Figure 1D, we selected some of the most active library members observed, and then determined that the sequences were quite similar to each other. Therefore we did not expect to find additional sequences that were much more active. We made a strategic decision to stop sequencing at four strong positives so that we could focus our efforts at exploring those four in as much detail as possible.

Comment: 3. Page 5, second paragraph: The authors state that they did not observe any long-term effect on cell viability. What do they mean with "long-term"? Please specify.

OUR RESPONSE: We intended "long term" to denote 24-48 hours in contrast to "acute" lysis (1-15 minutes) that some membrane-active peptides cause. However, we realize that this term is ambiguous and we have replaced it with more specific phrases such as "Cytotoxicity assays show no acute LDH leakage and no reduction metabolic potential or total protein in 24 hours at concentrations $\leq 40 \mu\text{M}$ " (now Page 6).

Comment: 4. Figure 3: SD seem to be relatively high, especially for conditions 21/37°C. Why is this the case? This observation should be discussed in the text. Did you test higher concentrations? May be this is the reason. Later on the authors use concentrations >10 μM at least.

OUR RESPONSE: We don't know exactly why TAMRA uptake is so variable, but we presume that delivery of PDEP-TAMRA conjugates must be especially sensitive to one or more experimental detail. We note that delivery of PNA (Fig. 2) and delivery of the GFP11 peptide (Fig. 4) are less variable. We did not explore higher concentrations in detail because we wanted to focus on the more useful low concentration range.

Comment: 5. Page 15: The authors should also include other library screens on CPPs within their discussion (e.g. Lin et al. *Angewandte Chem Int Ed* 2015, 54, 10370).

OUR RESPONSE: We thank the reviewer for reminding us of this work. On page 8/9 of the revised manuscript we now mention screening as well as computer-aided design papers in the literature, including the one mentioned here.

Reviewer #3 (Remarks to the Author):

The authors have a solid experience in synthetic molecular evolution (SME) of peptides. They now apply this interesting strategy to the synthesis of CPP-PNA libraries which has never been done to our knowledge.

Comment: Several publications indeed demonstrated that the Tat and Penetratin CPPs were poorly efficient to promote the delivery of antisense oligonucleotides analogues (such as PNA or PMO) geared at pre-mRNA splicing redirection. This has been demonstrated in the easy-to-implement HeLa705 splice correction assay and more recently in DMD (Duchenne muscular dystrophy) cells *in vitro* and *in vivo* using mdx mice.

Comment: In their short review of the literature, the authors seem to ignore that recent studies (by the groups of H.Moulton, M.Gait and others) have led to the discovery of new CPPs with a much higher delivery potential of conjugated PNAs (or more frequently PMOs) using these assays. Discussing these CPPs and comparing their efficiencies with the author's PDEPs should be done at least in the HeLa 705 assay. Comparing dose-response curves for selected PDEPs and control CPPs would help appreciating the claimed superiority of PDEPs. Comparison with oligoArg-based constructs indicates a modest (around 3x) increase in efficiency.

OUR RESPONSE: We thank the reviewer for reminding us of the work of these groups and others on phosphorodiamidate (antisense) morpholino oligomer (PMO/AMO) delivery by CPPs. Especially the important work on antisense PMO delivery in animal models of Duchenne MD and ALS. In the revised manuscript, we now discuss this body of work and cite papers from these authors in the introduction and in the discussion to put our work in a more complete, up to date context.

We were especially intrigued by the work of Iversen, Moulton, Gatti and colleagues (for example: "Arginine-rich cell-penetrating peptide dramatically enhances AMO-mediated ATM aberrant splicing correction and enables delivery to brain and cerebellum" *Human Molecular Genetics*, 2011, Vol. 20, No. 16 3151–3160). So we conducted a small head-to-head comparison of one of our peptides (P17) with the favored "peptide B" described in that paper and other papers. PepB enables delivery of AMOs with high efficiency, even *in vivo*, including delivery to the brain. Peptide B has the sequence RXRRBR-RBRRXR-linker- where X is 6-amino hexanoate and B is beta alanine. Thus Peptide B is essentially an Arg8 oligomer with longer (3 methylene) and shorter (1 methylene) linear spacers in the sequence.

We synthesized PepB-PNA705 and measured delivery in cell culture using the same approach described in the paper. We compared delivery to P17-PNA705, tat-PNA705, and no treatment. PepB is much more efficient than tat, which is only slightly more efficient than blank. Most importantly, PDEP P17 (which is the third least efficient of the PDEPs) was always more efficient

than PepB under all conditions we have studied, with a relative efficiency 5-20 fold greater. Of course we have not compared them *in vivo*, and we have not determined if PDEPs can deliver PMO efficiently.

These experiments have inspired us to begin a large parallel study of multiple families of CPPs (including our own) conjugated both to PNA and to PMOs to measure and compare PNA/PMO delivery efficiencies. It will be fascinating to determine if PNA delivery peptides efficiently deliver PMOs (or vice versa). Alternately it is possible that a new generation of synthetic molecular evolution will be required to identify maximally efficient PMO-delivery peptides. Data on Peptide B will be included when we publish that comparison. Therefore, we would prefer not to explicitly describe this preliminary comparison in this current paper, but instead publish a comprehensive parallel comparison when we have made all the measurements. For the reviewer, we include the new data below.

In the current paper we have re-examined the language in the sections describing the efficiency of the PDEPs to ensure that we do not claim superiority over peptides we have not actually tested.

Comment: Supplementary material does show higher toxicity of the PDEP-PNA conjugates as opposed to the parent Tat and Antp CPPs. This could be due to the increased Arg content of PDEPs and might become a problem for further clinical implementation. Such problems are considered as serious ones for the potential use of CPPs in DMD treatment.

OUR RESPONSE: We think the toxicity of the PDEP-PNA conjugates may be due, in part, to the PNA itself, coupled with especially efficient delivery when it is conjugated to PDEPS. This would

explain why the PDEP-PNA conjugates show higher toxicity (Supplementary Fig. 1) than the PDEPs alone (Supplementary Fig. 2) or PDEP-GFP11 conjugates (Supplementary figure 3).

Comment: The chloroquine (CQ) experiments should in principle be indicative of entrapment in endocytic vesicles. Somewhat puzzling however is the fact that CQ does not have any effect on oligoArg, Tat or Pen based conjugates. Fluorescence microscopy experiments with fluorochrome-tagged PNAs should help defining the efficiency with which PNAs are delivered to cell nuclei.

OUR RESPONSE: We were also puzzled by this observation. We believe that the lack of an effect of CQ on these other peptides may be due to the fact that we are testing much lower concentrations of peptides (5 μ M or less) compared to most studies in the literature (10-50 μ M). We are planning to conduct detailed studies on the mechanism of entry of these peptides, including the ones suggested above, but we feel that this story is too long and complex to add to this current paper.

Comment: At variance with most recent data in the literature for several CPPs, the uptake of PDEP-TAMRA conjugates is poorly affected by low temperature incubation in line with a direct translocation across the plasma membrane. This is somewhat in contradiction with CQ data and with the kinetics of cell uptake. Studies of cell trafficking should be done with fluorochrome-tagged CPP-ON conjugates and not with labelled CPPs as done here. It is well known indeed that the CPP cargo influences the overall behavior of the constructs. Accordingly, FACS and confocal microscopy experiments with fluorochrome-tagged PDEP-PNA should be done and would help discriminating between direct membrane translocation and endocytosis.

OUR RESPONSE: As we have stated in the revised paper in several locations (and also in a recent review Trends Biochem Sci. 2015 Dec;40(12):749-764. doi: 10.1016/j.tibs.2015.10.004) we believe that multiple mechanisms occur simultaneously: direct passage across the plasma membrane and endocytosis-dependent uptake and escape. Various experimental details, including peptide concentration, cargo, cell type and many other factors, affect how much each mechanism contributes. Various experimental techniques accentuate one mechanism or another. So, we conclude (based on low temperature entry) that direct entry occurs significantly, but we also conclude (due to the modest temperature and CQ effect) that endosomal uptake occurs to some extent. I am not sure that tracking dye labelled PDEP-PNA conjugates will provide resolution to these mechanistic questions. Such studies are often ambiguous because both mechanisms operate simultaneously. Such ambiguity can be plainly observed in the confocal microscopy images in Figure 3A.

Comment: In the last part, the authors investigated the potential of their peptides to deliver a 16-mer GFP1 peptide using the split-GFP assay. Again increased efficiency was achieved with PDEPs as compared to Tat, Pen or Arg although very high concentrations had to be used.

OUR RESPONSE: As we state in the manuscript on page 7, the delivery of the GFP peptide is a stringent assay (compared to PNA delivery) because it is stoichiometric, without amplification. One molecule delivered gives one unit of fluorescence. Therefore, we consider the GFP11 delivery at high concentration to be a reasonable demonstration of efficient delivery. In the near future, we

plan to screen a new library for maximally efficient delivery of GFP11 to determine if a different family of CPPs can delivery GFP11 efficiently at lower concentrations.

Comment: Additional experiments and a clear demonstration of increased efficiency as compared with recently described CPPs needs to be done before acceptance.

OUR RESPONSE: Please see our response above.

REVIEWERS' COMMENTS:

Reviewer #2 (Remarks to the Author):

The authors carefully revised their manuscript according to the reviewers comments. In my opinion, the paper is now ready for publication.

Reviewer #3 (Remarks to the Author):

I suggested the authors to provide a comparison of their new peptides with at CPPS (other than tat) proposed by other groups for PNA delivery. They have produced a very convincing set of data showing that their P17 peptide is significantly more efficient than Tat and importantly than peptide B which was proposed by the H.Moulton group as a very efficient PNA delivery vector. This would strongly improve the present manuscript and I therefore recommend to include these data in a revised version better than saving them for a further publications.

No clear answers could be provided to my questions concerning toxicity and trafficking of these new PNA conjugates. Although important issues, they might be the object of further studies which do not need to be included in the revised version.

REVIEWERS' COMMENTS:

Reviewer #2 (Remarks to the Author):

The authors carefully revised their manuscript according to the reviewers comments. In my opinion, the paper is now ready for publication.

Reviewer #3 (Remarks to the Author):

I suggested the authors to provide a comparison of their new peptides with at CPPS (other than tat) proposed by other groups for PNA delivery. They have produced a very

convincing set of data showing that their P17 peptide is significantly more efficient than Tat and importantly than peptide B which was proposed by the H.Moulton group as a very efficient PNA delivery vector. This would strongly improve the present manuscript and I therefore recommend to include these data in a revised version better than saving them for a further publications.

OUR RESPONSE: We have added the comparison data to Figure 2A in the paper and have added a small paragraph describing this result.

No clear answers could be provided to my questions concerning toxicity and trafficking of these new PNA conjugates. Although important issues, they might be the object of further studies which do not need to be included in the revised version.

OUR RESPONSE: Indeed they will be the subject of future studies.